# Assessment of the DNA Mismatch Repair System Is Crucial in Colorectal Cancers Necessitating Adjuvant Treatment: A Propensity Score-Matched and Win Ratio Analysis

**DOI:** 10.3390/cancers16010134

**Published:** 2023-12-27

**Authors:** Eva Lieto, Francesca Cardella, Duolao Wang, Andrea Ronchi, Giovanni Del Sorbo, Iacopo Panarese, Francesca Ferraraccio, Ferdinando De Vita, Gennaro Galizia, Annamaria Auricchio

**Affiliations:** 1Division of GI Tract Surgical Oncology, Department of Translational Medical Sciences, University of Campania “Luigi Vanvitelli”, 80138 Naples, Italy; eva.lieto@unicampania.it (E.L.); francesca.cardella@unicampania.it (F.C.); giovanni.delsorbo@studenti.unicampania.it (G.D.S.); annamaria.auricchio@unicampania.it (A.A.); 2Department of Clinical Sciences, School of Tropical Medicine Liverpool, Liverpool L7 8XZ, UK; duolao.wang@lshtm.ac.uk; 3Division of Pathology, Department of Mental and Physical Health and Rehabilitation Medicine, University of Campania “L VanvItelli”, via Luciano Armanni, 80138 Naples, Italy; andrea.ronchi@unicampania.it (A.R.); iacopo.panarese@gmail.com (I.P.); franca.ferraraccio@unicampania.it (F.F.); 4Division of Medical Oncology, Department of Precision Medicine, School of Medicine, University of Campania ‘Luigi Vanvitelli’, 80138 Naples, Italy; ferdinando.devita@unicampania.it

**Keywords:** colorectal cancers, DNA mismatch repair system, immunotherapy, win ratio

## Abstract

**Simple Summary:**

Around 15% of sporadic colorectal cancers exhibit microsatellite instability. This unique tumor population appears to be poorly responsive to conventional chemotherapy and, conversely, reveals excellent results with immunotherapy. Our data, as demonstrated by propensity score-matched and win ratio analyses, show that there are no substantial differences between unstable and stable tumors in early colorectal cancers treated with surgery alone. On the contrary, stable tumors did much better than unstable tumors in the advanced stages of colorectal cancer undergoing conventional adjuvant treatment. Determination of the status of the DNA mismatch repair system is crucial in high-risk colorectal cancers to optimize treatment.

**Abstract:**

A deficient DNA mismatch repair (MMR) system is identified in a non-negligible part of sporadic colorectal cancers (CRCs), and its prognostic value remains controversial. High tumor mutational burden, along with a poor response to conventional chemotherapy and excellent results from immunotherapy, are the main features of this subset. The aim of this study was to evaluate the predictive value of DNA MMR system status for its best treatment. Four hundred and three CRC patients, operated on from 2014 to 2021 and not treated with immunotherapy, entered this study. Immunohistochemistry and polymerase chain reaction, as appropriate, were used to unequivocally group specimens into microsatellite stable (MSS) and instable (MSI) tumors. The win-ratio approach was utilized to compare composite outcomes. MSI tumors accounted for 12.9% of all series. The right tumor location represented the most important factor related to MSI. The status of the DNA MMR system did not appear to correlate with outcome in early-stage CRCs not requiring adjuvant treatment; in advanced stages undergoing conventional chemotherapy, MSI tumors showed significantly poorer overall and disease-free survival rates and the highest win ratio instead. The determination of DNA MMR status is crucial to recommending correct management. There is clear evidence that instable CRCs needing adjuvant therapy should undergo appropriate treatments.

## 1. Introduction

Colorectal cancer (CRC) is one of the most prevalent malignancies worldwide [1]. Currently, CRC is believed to be not a single disease but a galaxy of different subtypes resulting from combinations of genetic events and epigenetic alterations [2]. In sporadic CRC, chromosomal instability is believed to promote carcinogenesis in about 85% of cases. The remaining 15% of cases appear to be correlated with alterations in the DNA mismatch repair (DNA MMR) system [3]. This system seems to play a crucial role in the suppression of mutagenesis and cancer; during DNA replication, this condition leads to uncorrect base mispairs with small insertion-deletion mutations, resulting in an elevated tumor mutational burden and slippage at repetitive DNA microsatellites, the so-called microsatellite instability [4]. Differences in clinico-pathological features and their prognostic role between microsatellite stable and unstable CRCs have been extensively investigated with conflicting results [5]. However, the most important issue is that a defective DNA MMR system leads to the production of aberrant peptides and proteins, which cause a strong immune reaction with tumor infiltration by cytotoxic T-cell lymphocytes due to their immunogenic neoantigens [6]. Theoretically, this condition should promote tumor control, but the fact that all such cancers usually grow demonstrates that immune escape consistently occurs [4]. Since recent studies have emphasized the concomitant expression of multiple active immune checkpoint markers, able to induce the functional exhaustion and unresponsiveness of T cells, the use of immune checkpoint inhibitors (ICIs) may appear as a rational strategy in this setting [7]. Consistently, immunotherapy with ICIs shows to yield excellent responses in such tumors, both in the neo- and adjuvant setting, and, since the uncertain response of microsatellite instable CRCs to conventional chemotherapy [2,8,9], it might dramatically change the scenario of therapeutic strategies [10,11,12,13,14,15,16].

In CRC, most studies focused on clinical outcome have analyzed each prognostic factor as a single independent parameter. More recently, composite metrics to assess patient outcomes have been increasingly adopted [17]. A composite endpoint including two (or more) types of clinical events enables a more holistic comparison of two different conditions. However, some criticisms have been raised since this method ignores the differences in clinical severity of the individual components [18]. Therefore, the win ratio (WR) method, which both takes into consideration the proportion of patients achieving each component outcome and orders outcomes in their hierarchical importance, has been proposed [19].

The aim of this study was to assess, in a large cohort of CRC patients with homogenous surgical and oncological treatments, the prognostic and predictive value of the DNA MMR system status. In addition, we defined the odds of achieving a winner for patients grouped into stable and unstable tumors.

## 2. Material and Methods

### 2.1. Patients

A cohort of 513 patients with colorectal cancer undergoing surgery from 1 January 2014, to 31 December 2021, were initially retrieved from our own prospectively maintained database. Missing data in the years prior to 2014 conditioned the start of the analysis; in addition, the end of this study were chosen both to ensure at least one year of follow-up time and because of a radical change of the treatment protocol at the beginning of 2022, according to the new data from the literature, in order to avoid analyzing non-homogeneous patients with possible misinterpretation of the results.

Suspected or confirmed Lynch’s syndrome (12 patients) and rectal cancers (98 patients), which are known to modify tumor environments, were primarily excluded from this study [20]. Finally, 403 patients entered this study (Figure 1). For each patient, the following data were collected: age, gender, tumor, basal carcinoembryonic antigen serum level, performance status according to the Easthern Cooperative Oncology Group (ECOG) scale, preoperative Naples Prognostic Score [21], tumor size, tumor-node-metastasis stage [22], Dukes staging system, histologic differentiation, surgical radicality, postoperative morbidity, postoperative chemotherapy, recurrence rate, overall (OS) and disease-free survival (DFS) rates, and stability of the DNA MMR system. No patient in this series underwent neoadjuvant treatment. Complete clinico-pathological data were available for all patients, including long-term outcomes. Patients discharged from the hospital underwent oncological counseling and treatment as appropriate (particularly for radically resected patients: adjuvant therapy for intermediate-high-risk stage II, as well as for stages III and IV; for non-radically resected patients: first-line chemotherapy, including biological drugs in metastatic patients) [8]. No patient was lost to follow-up that was completed by 31 March 2023. All patients gave their informed consent. This study was approved by the Ethical Committee of the Department of Translational Medical Sciences of the University of Naples, ‘Luigi Vanvitelli’ (Ethical Committee n° 6 of DTMS, Naples, Italy, on 9 January 2023). This study was registered at ClinicalTrials.gov (identifier: NCT05871567).

### 2.2. Immunohistochemistry and Polymerase Chain Reaction

Formalin-fixed and paraffin-embedded 4 µm-thick whole tissue sections were cut and used for immunohistochemical analysis. Immunohistochemistry (IHC) was carried out using the following antibodies: MLH1 (clone ES05), MSH2 (clone G219-1129), MSH6 (clone EPR3945), and PMS2 (clone A16-4). The immunohistochemical slides were interpreted by two different pathologists, who blinded each other and had no prior knowledge of clinicopathological parameters. In addition, pathology reports were reviewed by a third pathologist. Discrepancies between investigators (<5% of the cases) required a further joint observation with conclusive agreement. MMR protein loss was defined by the absence of immunohistochemical staining in the nucleus of the neoplastic cells. Three different classes of MMR protein expression were defined: (i) proficient MMR (pMMR): cases with expression of all four MMR proteins; (ii) deficient MMR (dMMR): cases with absent expression of one of two heterodimers, including MLH1/PMS2 and/or MSH2/MSH6; (iii) reduced MMR (low-patchy MMR): cases with loss of one MMR protein and/or the heterogeneous/patchy expression of one or more MMR proteins. Heterogeneous expression was defined as strongly immunoreactive cells admixed with negative cells, and patchy expression was defined as confluent areas of staining loss. In these cases, as previously described, additional molecular analysis by polymerase chain reaction (PCR) was used to unequivocally categorize cases into microsatellite stable (MSS) and microsatellite unstable (MSI) tumors [23]. MSI was determined on tumor DNA using EasyPGX^®^, including the following mononucleotide repeats: BAT25, BAT26, NR21, NR22, NR24, NR27, CAT25, and MONO27.

From now on, microsatellite instability/DNA mismatch repair deficiency (MSI/dMMR) tumors will be referred to as MSI tumors, and microsatellite stability/proficient mismatch repair (MSS/pMMR) tumors will be referred to as MSS tumors.

### 2.3. Statistical Analysis

Statistical analysis was carried out using the SPSS 21.0 software (SPSS Inc., Chicago, IL, USA) and the statistical package R (version 4.2.3). All statistical tests were performed two-sidedly, and a *p* value less than 0.05 was considered to be statistically significant. The comparisons between proportions were made by using the chi-square test. Multiple regression was used to investigate correlations between DNA MMR status and other factors. To accurately investigate prognostic factors of oncological outcome, the statistical analyses related to overall and disease-free survival rates were restricted only to patients undergoing potentially curative surgery. Univariate statistical analysis was determined by a log-rank test; curves were plotted using the Kaplan–Meier method, and *p* values and hazard ratios (HR) with a 95% confidence interval (CI) were obtained. The independent significance of each factor was determined by Cox’s proportional hazards model, following the inclusion of prognostic variables showing a significant *p* value on univariate analysis. The model of the best linear combination of variables able to predict OS and DFS rate (serum CEA levels and tumor stage) along with factors directly correlated with the status of the DNA MMR system (namely, tumor location and adjuvant chemotherapy) was introduced in a multivariate logistic regression to calculate a propensity score for each patient. Then, a propensity score-matched analysis was performed to re-evaluate univariate and multivariate analyses in the matched couples [24,25]. Finally, to obtain an estimate of a composite endpoint accounting for clinical priorities, namely overall survival and disease-free survival, a win ratio analysis using an unmatched approach was performed, providing an informative estimate of the different statuses of the DNA MMR system with a 95% CI and *p* value in colorectal cancer patients. This method has been extensively described elsewhere [18,19]. In brief, to calculate the WR, every patient with an MSI tumor was compared with every patient with an MSS tumor. In each comparison, the major event (death) and time from surgical operation to event or last follow-up (censored) were assessed. If one patient died first, the other patient won. Otherwise, the second event (tumor relapse) was assessed using the same principles. The pairs that did not fulfill the above criteria were defined as tied or no winners. The ratio between winners and losers represented the win ratio; it could be interpreted as the odds that one group had of winning over the other group (better survival and lower recurrence rate). In addition, we also calculated the net benefit, which, among all pairwise comparisons, indicated the percentage with which one group had a better win ratio than the other group, and the win odds, which indicated the odds of a group winning over the other group [26].

## 3. Results

Clinico-pathological characteristics of the series are listed in Table 1. At IHC analysis, 318 (78.9%) tumors were judged as pMMR, thirty-three (8.2%) as dMMR, and fifty-two (12.9%) as low-patchy. The latter were submitted to PCR analysis, and nineteen tumors were classified as MSI and thirty-three as MSS tumors. Overall, 351 tumors (87.1%) were judged as pMMR-MSS cancers, while fifty-two tumors (12.9%) were judged as dMMR-MSI cancers. At univariate analysis, MSI tumor rate correlated with the right location of the tumor (*p* = 0.005); at multiple regression analysis, MSI tumors were shown to be related to higher rates of right location (*p* = 0.01), metastatic disease (*p* = 0.02), and no radical resection (*p* = 0.001).

### 3.1. Survival and Tumor Recurrence Analysis

As specified above, survival analysis was restricted to patients undergoing radical surgery. At the time when the 339 patients were subjected to the analysis, 72 (21.2%) patients had died (39 for tumor relapse and 33 for other causes than colorectal cancer), and 267 patients were alive (251 patients without tumor recurrence). The tumor recurrence rate was 16% (55 patients). The 1 to 5-year OS rates were 95.8, 90.5, 85.4, 79.7, and 75.7%, respectively. Instead, the 1–5-year DFS rates were 93.7, 88.5, 83.8, 81.0, and 80.0%, respectively. Table 2 shows univariate analyses related to overall and disease-free survival rates. Although MSS cancers appeared to perform significantly better than MSI tumors, at multivariate analysis, the status of DNA MMR lost its significance, showing only a weak value (*p* = 0.06) for DFS (Table 3A). It was supposed to be related to the poor response to conventional adjuvant chemotherapy observed in MSI cancers. Indeed, among 182 patients with no need for adjuvant chemotherapy (low-risk TNM stage ≤ II), no significant differences in OS and DFS rates were seen between MSS and MSI patients, respectively. On the contrary, among 157 patients undergoing adjuvant chemotherapy (a heterogeneous group of node-positive and/or pT4 and/or radically resected liver metastases patients; i.e., high-risk stage II patients and stage ≥ III cancers), MSS patients showed significantly better DFS rates (HR = 0.37; 95% CI 0.14–0.89; *p* = 0.005) and nearly significant OS rates (HR = 0.51; 95% CI 0.20–1.29; *p* = 0.07) than MSI patients, thus demonstrating that MSI tumors in advanced stages, requiring adjuvant therapy, are significantly disadvantaged or even harmed with conventional chemotherapy schedules (Figure 2 and Figure 3).

### 3.2. Propensity Score-Matched Comparison

A propensity score-matched analysis was applied to further support the results observed in the whole series.

Forty-six MSI tumors were matched, in a 1:2 fashion, with ninety-two MSS tumors. The overall χ^2^ balance test was not significant (*p* = 0.96), and the λ_1_ measure was larger in the unmatched sample (0.288) than in the matched sample (0.130), indicating improved overall balance with matching [25]. On univariate analysis, DNA MMR status was shown to be even more significant [5-year OS rates were 81.6 and 60.8% (HR = 0.43; 95%CI 0.19–0.97; *p* = 0.02), and 5-year DFS rates were 86.3 and 70.5% (HR = 034; 95%CI 0.13–0.90; *p* = 0.01) in MSS and MSI tumors, respectively]. On multivariate analysis, DNA MMR was confirmed to be an independent prognostic factor of tumor recurrence and did better than the whole series concerning overall survival. Interestingly, adjuvant chemotherapy, once again, was not selected as an independent prognostic factor; other covariates, such as tumor stage and DNA MMR status, affected the oncological outcome (Table 3B).

**Table 3 cancers-16-00134-t003:** Multivariate analysis related to overall and disease-free survival rates in: (A) 339 patients undergoing radical resection. (B) Approximately 138 patients selected by propensity-score matched analysis by using the 1:2 nearest neighbor technique with a small caliper of 0.15. Cox’s proportional-hazards model.

A					
Overall Survival					
	Coefficient	Standard Error	Hazard Ratio	95% CI ^a^ Hazard Ratio	*p* Value
Age (years)	1.0349	0.2857	2.81	1.61–4.91	**0.0003**
Serum CEA Levels	0.9318	0.2647	2.53	1.51–4.25	**0.0004**
Performance Status	0.5203	0.2127	1.68	1.11–2.54	**0.01**
NPS ^b^	0.5136	0.2069	1.67	1.11–2.50	**0.01**
pT Stage	0.1050	0.1798	1.11	0.78–1.57	0.55
pN Stage	0.0920	0.0169	1.09	1.06–1.13	**0.0001**
pM Stage	0.0351	0.0299	1.03	0.97–1.09	0.24
Postoperative Complications	1.0105	0.3331	2.74	1.43–5.25	**0.002**
Microsatellite Status	0.3322	0.3237	1.39	0.74–2.62	0.30
**Disease-Free Survival**					
	**Coefficient**	**Standard Error**	**Hazard Ratio**	**95% CI ^a^** **Hazard Ratio**	***p* Value**
Serum CEA Levels	0.7871	0.2898	2.19	1.24–3.86	**0.006**
NPS ^b^	0.2786	0.2271	1.32	0.84–2.05	0.21
pT Stage	0.6840	0.2386	1.98	1.24–3.15	**0.004**
pN Stage	0.0647	0.0212	1.06	1.02–1.11	**0.002**
pM Stage	0.0809	0.0303	1.08	1.02–1.15	**0.007**
Adjuvant Chemotherapy	0.3646	0.4156	1.43	0.64–3.23	0.38
Microsatellite Status	0.6403	0.3461	1.89	0.96–3.72	0.06
**B**					
**Overall Survival**					
	**Coefficient**	**Standard Error**	**Hazard Ratio**	**95% CI ^a^** **Hazard Ratio**	***p* Value**
Age (years)	0.8943	0.4485	2.44	1.01–5.86	**0.04**
Serum CEA Levels	0.9434	0.4884	2.56	0.99–6.65	0.05
Performance Status	0.7139	0.3507	2.04	1.03–4.04	**0.04**
NPS ^b^	0.4568	0.3472	1.57	0.80–3.10	0.18
pT Stage	0.2364	0.3345	1.26	0.65–2.43	0.47
pN Stage	0.0999	0.0275	1.10	1.04–1.16	**0.0003**
pM Stage	0.0157	0.0499	1.01	0.92–1.11	0.75
Postoperative Complications	1.2384	0.5650	3.45	1.14–10.38	**0.02**
Microsatellite Status	0.6328	0.4451	1.88	0.79–4.48	0.15
**Disease-Free Survival**					
	**Coefficient**	**Standard Error**	**Hazard Ratio**	**95% CI ^a^** **Hazard Ratio**	***p* Value**
Serum CEA Levels	0.4538	0.4975	1.57	0.59–4.15	0.36
NPS ^b^	0.4640	0.3955	1.59	0.73–3.43	0.24
pT Stage	0.8357	0.3939	2.30	1.07–4.97	**0.03**
pN Stage	0.0505	0.5088	1.05	0.39–2.83	0.92
pM Stage	0.1006	0.0477	1.10	1.00–1.21	**0.03**
Adjuvant Chemotherapy	0.6785	0.7837	1.97	0.42–9.08	0.38
Microsatellite Status	1.1049	0.4771	3.01	1.19–7.66	**0.02**

A multivariate analysis was performed, including variables with significant values on the univariate. Analysis: Postoperative complications were defined as grade II or higher of the Clavien-Dindo classification [27]. ^a^ 95% confidence interval. ^b^ Naples Prognostic Score (reference n° 21).

### 3.3. Win Ratio Analysis

When analyzing all patients (18,252 pairwise comparisons) and patients undergoing non-radical resection (348 pairwise comparisons), the win ratio failed to recognize significant differences between the two groups. On the contrary, in cancers undergoing radical resection (13,478 pairwise comparisons), patients with MSS tumors were shown to have a significantly better oncological outcome than patients with MSI tumors, with a win ratio = 2.04, a 16% oncological benefit, and a 38% probability of having better results. However, in early-stage CRCs not requiring adjuvant chemotherapy, no differences were seen between MSS and MSI tumors. Interestingly, MSI tumors were associated with a significant disadvantage only in advanced CRCs treated with conventional adjuvant chemotherapy (WR = 2.47). These effects were confirmed in the stratified subgroup of patients selected by the propensity score-matched analysis controlled for confounding factors and even improved when selecting tumor recurrence as the primary event. In this setting, MSS tumors displayed a win ratio of 2.61 and 2.76, 20% and 21% oncological benefit, and 49% and 53% of odds having a better result than MSI tumors, respectively (Table 4).

## 4. Discussion

This study confirms that the incidence of MSI in sporadic CRCs is not negligible and requires careful investigation for its implications on prognosis and treatment. These tumors arise from a deficiency of the DNA MMR system, which leads to multiple errors in repetitive DNA sequences (called microsatellites). This is an important cause for genomic instability and cell proliferation up to the point of colorectal cancer generation [28]. Its evaluation has been recently defined by a guideline from the College of American Pathologists, endorsed by the ASCO, that recommended the use of MMR-IHC and/or MSI by PCR for the detection of DNA MMR defects in CRCs [29,30].

In this study, we applied an even more rigorous method, consisting of both double-blind IHC analysis and confirmation of doubtful cases (heterogeneous/patchy expression) by PCR, in order to be as confident as possible about the results [23].

In the last decade, several studies have investigated the prognostic and predictive value of the MSI phenotype in CRCs, with inconclusive and sometimes contradictory results [31,32]. Initial observations of MSI status as a favorable and unfavorable molecular prognostic marker in localized and metastatic CRCs, respectively, have been questioned by more recent investigations [5]. This is mainly due to the limited number of MSI tumors, as compared to MSS cancers, and the inclusion of patients with potential differences in study time frames, adjuvant regimens, and tumor stages. To overcome these issues, we analyzed a homogenous group of CRC patients treated by the same surgical and oncological team, undergoing conventional chemotherapy and no immunotherapy. In addition, the limited number of MSI tumors, when compared to larger databases, were addressed by using adequate statistical methods.

Because of their peculiar immunologic characteristics, MSI CRCs have been suggested to be more frequent in the early stages [4,33], a finding that was not confirmed in this study; actually, an even higher percentage of metastatic disease were MSI tumors. However, MSI cancers were uniformly distributed across tumor stages, which makes our results even more accurate. Overall, MSI tumors displayed poorer OS and DFS rates than MSS tumors, but, not surprisingly, the initial multivariate analysis failed to assign independence to this covariate. The DNA MMR status often lacks its predictive power because of the short number of MSI tumors to be compared with other more common prognostic factors (such as tumor stage), often inducing controversial results [34]. However, after grouping patients according to tumor stages and adjuvant chemotherapy, the DNA MMR system status did not appear to modify the outcome in patients with early-stage CRCs requiring surgery alone without any adjuvant treatment. On the contrary, in patients with advanced CRC stages and therefore undergoing conventional adjuvant chemotherapy, MSS tumors were associated with significantly better OS and DFS rates than MSI tumors, as confirmed by propensity score-matched analysis. In addition, the win ratio approach, based on more holistic and hierarchical composite metrics, dispelled all doubts by showing that MSS and MSI tumors apparently equalized in the whole series; conversely, in radically resected patients, MSS tumors had a better outcome, albeit only in patients undergoing conventional adjuvant chemotherapy. Interestingly, higher win ratio values were obtained by reversing the composite endpoints, suggesting that the status of the DNA MMR system was particularly correlated with tumor recurrence. To the best of our knowledge, this is the first time that the win ratio approach, along with propensity score-matched analysis, has been applied to explore the status of the DNA MMR system in colorectal cancers [17].

An intriguing aspect of CRC biology is the MSI-driven high tumor mutational burden, which leads to the expression of immunogenic neoantigens able to elicit brisk immune responses that appear nonetheless insufficient [6,35]. Therefore, restoring effective anti-tumor immune responses is the rationale for the implementation of ICI immunotherapy [36]. Along with the uncertain responsiveness of MSI CRCs to conventional chemotherapy, several recent studies have reported excellent results for immunotherapy of this unique CRC population, both in neo- and adjuvant settings [2,9]. These observations open new scenarios in the treatment of colorectal cancers, both in terms of sparing further treatments, including surgery and in the diffusion of immunotherapy once it has overcome the immune escape phenomenon and made tumor cells more sensitive to appropriate treatments [10,11,12,13,14,15,16,37].

This study had some limitations. Firstly, the single recruitment site and limited number of patients may propose results that are misleading and give wrong information. To minimize selection bias, we included consecutive patients treated consistently and analyzed them with statistical techniques, which suggests that we can be particularly confident in the results. Secondly, to overcome the interobserver variability among pathologists, we adopted a rigid protocol with three pathologists, further joint observation, and PCR-integrated analysis in low-patchy MMR cases. Lastly, although a multivariate regression analysis was performed, the observed results may still be subject to some unobserved confounding factors.

## 5. Conclusions

Our study shows no substantial differences between MSS and MSI tumors in early-stage CRCs that were surgically resected and not scheduled for further treatments. On the contrary, MSS tumors did much better than MSI tumors, with significantly different OS and DFS rates, in patients with advanced CRCs requiring adjuvant treatment. Therefore, assessment of DNA MMR system status is crucial to recommending correct management [8,38].

## Figures and Tables

**Figure 1 cancers-16-00134-f001:**
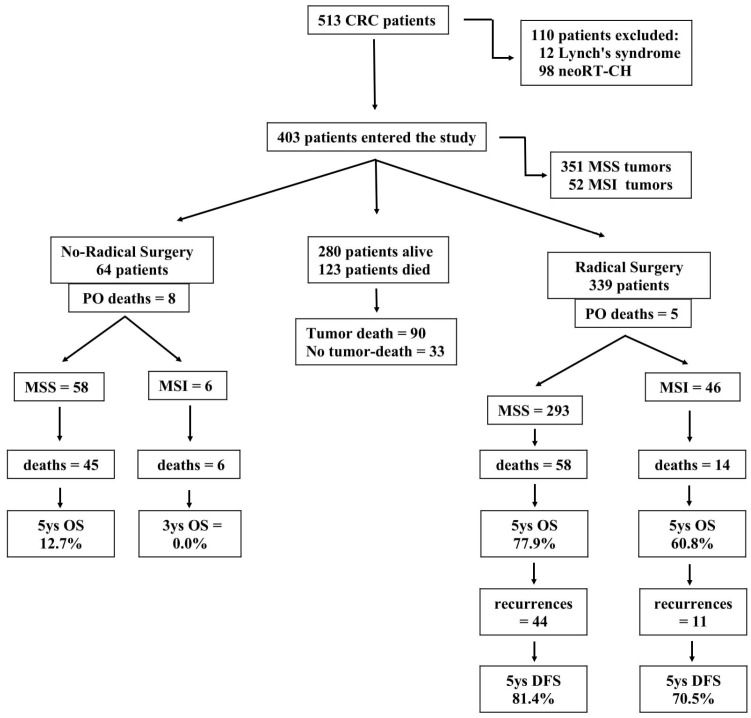
Flow chart of this study population.

**Figure 2 cancers-16-00134-f002:**
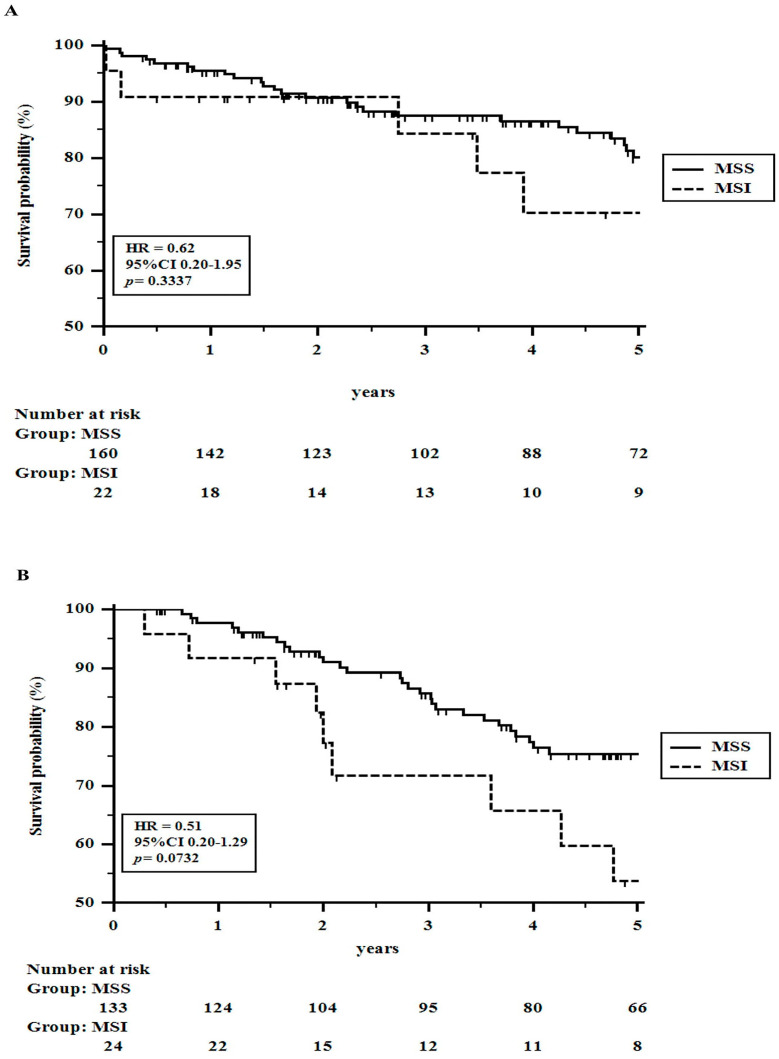
Approximately 1 to 5-years Overall Survival rates in colorectal cancer patients undergoing radical surgery stratified for adjuvant chemotherapy and the status of the DNA MMR system: (**A**) A total of 182 patients with early colorectal cancer are not undergoing adjuvant chemotherapy. (**B**) A total of 157 patients with advanced colorectal cancer undergoing adjuvant chemotherapy.

**Figure 3 cancers-16-00134-f003:**
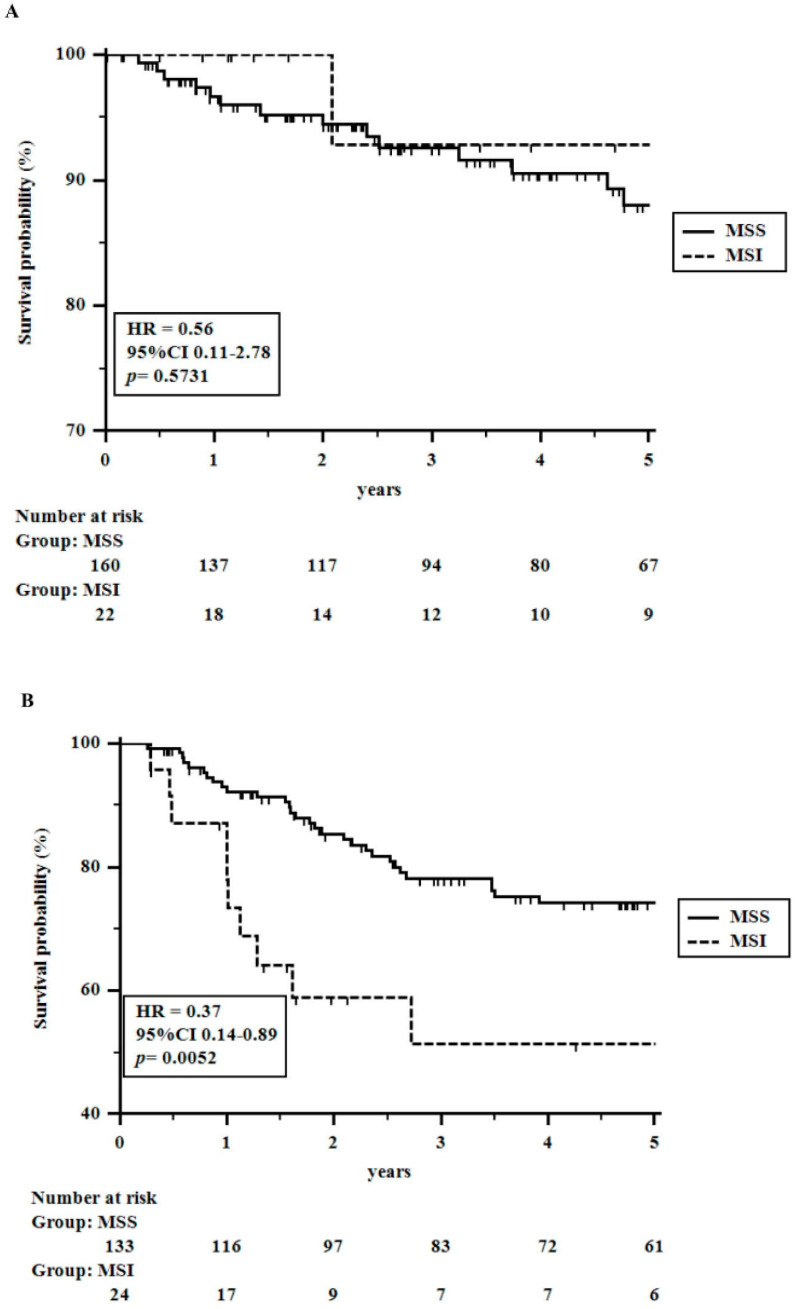
Approximately 1 to 5-year Disease-free Survival rates in colorectal cancer patients undergoing radical surgery stratified for adjuvant chemotherapy and the status of the DNA MMR system: (**A**) 182 patients with early colorectal cancer not undergoing adjuvant chemotherapy; (**B**) 157 patients with advanced colorectal cancer undergoing adjuvant chemotherapy.

**Table 1 cancers-16-00134-t001:** Clinico-pathological characteristics of the series and relations between microsatellite stable tumors (MSS) and microsatellite instable tumors (MSI).

	All (n = 403)	MSS (n = 351)	MSI (n = 52)	*p* Value ^a^
Age (years) ^b^				0.55
≤68 years	221	190	31	
>68 years	182	161	21	
Gender				0.34
Male	207	184	23	
Female	196	167	29	
Tumor Site ^c^				**0.005**
Right Colon	145	115	30	
Left Colon	145	134	11	
Rectum	113	102	11	
Serum CEA Levels				0.98
≤3.5 ng/mL	237	207	30	
>3.5 ng/mL	166	144	22	
Performance Status				0.69
0	120	102	18	
1	203	178	25	
2	80	71	9	
NPS ^d^				0.79
0	76	67	9	
1	211	185	26	
2	116	99	17	
Tumor Size ^b^				0.87
≤4 cm	240	208	32	
>4 cm	163	143	20	
pT Stage				0.48
0	9	8	1	
1	22	20	2	
2	82	73	9	
3	217	187	30	
4a	65	56	9	
4b	8	7	1	
1–2	113	101	12	0.49
3–4	290	250	40	
pN Stage				0.62
0	245	215	30	
1a	46	40	6	
1b	48	42	6	
1c	13	11	2	
2a	15	12	3	
2b	27	22	5	
Nx	9	9	0	
Node Negative	245	215	30	0.73
Node Positive	149	127	22	
pM Stage				0.08
0	314	278	36	
1a	66	55	11	
1b	10	8	2	
1c	13	10	3	
M−	314	278	36	0.15
M+	89	73	16	
TNM Stage				0.44
0–II	225	199	26	
III–IV	178	152	26	
Dukes’s Stage				0.21
A	99	88	11	
B	126	111	15	
C	89	79	10	
D	89	73	16	
Histologic Differentiation				0.26
well	8	8	0	
moderate	353	309	44	
poor	42	34	8	
Radical Resection				0.47
Yes	339	293	46	
No	64	58	6	
PostOperative Complications				0.31
No	361	317	44	
Yes	42	34	8	
Postoperative Chemotherapy				0.99
No	190	166	24	
Yes	213	185	28	
Tumor Recurrence ^e^				0.70
No	284	249	35	
Yes	119	102	17	
Survival				0.24
Yes	280	248	32	
No	123	103	20	

*p* value refers to MSS vs. MSI tumors; CEA indicates carcinoembryonic antigen (3.5 ng/mL normal value in healthy subjects); Postoperative complications were defined as grade II or higher of the Clavien-Dindo classification [27]. ^a^ chi-square test. ^b^ median value in all series. ^c^ right colon means cecum, the ascendent colon, and the right transverse colon; the left colon means left. transverse colon, descendent colon, and sigma. ^d^ Naples Prognostic Score (reference n° 21). ^e^ including 64 non-radical operations.

**Table 2 cancers-16-00134-t002:** Univariate analysis related to overall and disease-specific survival in 339 colorectal cancer patients undergoing potentially curative surgery.

		Overall Survival (OS)			Disease-Free Survival (DFS)		
	Nr.	5-Years Rate %	HR 95% CI ^a^	*p* Value	5-Years Rate %	HR 95% CI ^a^	*p* Value
Age (years) ^b^							
≤68 years	183	87.0	0.29	**<0.0001**	79.1	0.90	0.71
>68 years	156	61.3	0.18–0.46		81.0	0.13–1.54	
Gender							
Male	176	73.8	0.74	0.21	80.2	1.06	0.81
Female	163	77.7	0.46–1.18		79.8	0.62–1.80	
Tumor Site ^c^							
Right Colon	122	71.8			81.2		
Left Colon	120	75.8	//	0.25	80.4	//	0.79
Rectum	97	80.3			78.1		
Serum CEA Levels							
≤3.5 ng/mL	230	84.4	0.30	**<0.0001**	86.2	0.32	**<0.0001**
>3.5 ng/mL	109	57.0	0.18–0.50		64.9	0.17–0.58	
Performance Status							
0	113	88.6			83.8		
1	166	74.9	//	**<0.0001**	76.3	//	0.36
2	60	53.2			82.2		
NPS ^d^							
0	73	88.7			87.9		
1	176	80.9	//	**<0.0001**	82.7	//	**0.004**
2	90	54.1			63.2		
Tumor Size ^b^							
≤4 cm	217	76.8	0.82	0.42	81.8	0.68	0.15
>4 cm	122	73.9	0.50–1.34		76.8	0.38–1.19	
pT Stage							
0	8	100.0			100.0		
1	22	81.4			92.9		
2	81	87.1	//	**<0.0001**	90.6	//	**<0.0001**
3	185	72.5			78.1		
4a	40	62.7			57.6		
4b	3	0.0			33.3		
1–2	111	87.0	0.35	**0.0007**	91.9	0.25	**0.0003**
3–4	228	70.0	0.22–0.57		73.9	0.14–0.44	
pN Stage							
0	233	83.0			86.6		
1a	38	75.9			76.7		
1b	32	68.1	//	**<0.0001**	68.0	//	**<0.0001**
1c	11	23.9			54.5		
2a	9	64.8			62.5		
2b	16	27.7			36.6		
Node Negative	233	83.0	0.33	**<0.0001**	86.6	0.32	**<0.0001**
Node Positive	106	60.4	0.19–0.55		65.1	0.17–0.58	
pM Stage							
0	313	79.7			83.7		
1a	15	46.7	//	**<0.0001**	42.9	//	**<0.0001**
1b	4	25.9			50.0		
1c	7	14.3			16.7		
M−	313	79.7	0.21	**<0.0001**	83.7	0.15	**<0.0001**
M+	26	26.2	0.08–0.55		36.7	0.04–0.48	
TNM Stage							
0–II	225	83.8	0.31	**<0.0001**	88.0	0.26	**<0.0001**
III–IV	114	60.6	0.19–0.51		63.8	0.14–0.46	
Dukes’s Stage							
A	99	90.4			90.0		
B	126	81.3	//	**<0.0001**	85.6	//	**<0.0001**
C	88	69.7			72.4		
D	26	34.6			36.7		
Histologic Differentiation							
well	8	100.0			100.0		
moderate	308	76.3	//	0.14	79.1	//	0.43
poor	23	68.3			89.7		
PostOperative Complications							
No	310	77.2	0.32	**0.0001**	80.1	0.92	0.92
Yes	29	59.5	0.13–0.82		78.4	0.32–2.64	
Adjuvant Chemotherapy							
No	182	79.0	0.68	0.10	88.6	0.32	**<0.0001**
Yes	157	72.2	0.42–1.08		70.8	0.18–0.54	
Microsatellite Status							
MSS	293	77.9	0.54	**0.03**	81.4	0.50	**0.03**
MSI	46	60.8	0.26–0.82		70.5	0.21–0.77	

CEA indicates carcinoembryonic antigen (3.5 ng/mL is the normal value in healthy subjects). ^a^ Hazard rate with a 95% Confidence Interval; Postoperative complications were defined as grade II or higher of the Clavien-Dindo classification [27]. ^b^ median value in the series. ^c^ right colon means cecum, ascendent colon, and right transverse colon; the left colon means left transverse colon, descendent colon, and sigma. ^d^ Naples Prognostic Score (reference n° 21).

**Table 4 cancers-16-00134-t004:** Win Ratio Analysis.

	n° Pairwise Comparisons	Win Ratio 95%CI *p* Value	Net Benefit 95%CI *p* Value	Win Odds 95%CI *p* Value	% of Winning ^a^	Pairs Tied ^b^
All patients (403)		1.47	0.11	1.25		
MSS = 351	18,252	0.91–2.38	−0.03–0.25	0.94–1.65	MSS = 34	7646
MSI = 52		0.11	0.12	0.12	MSI = 23	
NonRadical (64)		1.34	0.14	1.34		
MSS = 58	348	0.39–4.53	−0.47–0.75	0.39–4.5	MSS = 57	3
MSI = 6		0.64	0.64	0.64	MSI = 42	
Radical (339)		2.04	0.16	1.38		
MSS = 293	13,478	1.10–3.79	0.02–0.30	1.03–1.84	MSS = 31	7207
MSI = 46		**0.02**	**0.02**	**0.02**	MSI = 15	
Radical ^c^ (339)		2.20	0.17	1.42		
MSS = 293	13,478	1.15–4.21	0.02–0.33	1.05–1.92	MSS = 32	7207
MSI = 46		**0.01**	**0.02**	**0.02**	MSI = 14	
Radical						
AdjCH = No						
(182)		1.37	0.05	1.10		
MSS = 160	3520	0.47–3.98	−0.12–0.21	0.79–1.53	MSS = 18	2432
MSI = 22		0.56	0.57	0.57	MSI = 13	
Radical						
AdjCH = Yes						
(157)		2.47	0.26	1.69		
MSS = 133	3192	1.16–5.23	0.03–0.49	1.07–2.67	MSS = 43	1254
MSI = 24		**0.01**	**0.02**	**0.02**	MSI = 17	
Propensity (138)		2.61	0.20	1.49		
MSS = 92	4232	1.27–5.37	0.04–0.36	1.08–2.06	MSS = 32	2.349
MSI = 46		**0.009**	**0.01**	**0.01**	MSI = 12	
Propensity ^c^ (138)		2.76	0.21	1.53		
MSS = 92	4232	1.32–5.78	0.04–0.37	1.10–2.12	MSS = 33	2.349
MSI = 46		**0.007**	**0.01**	**0.01**	MSI = 12	

Win Ratio: ratio between winners and losers in all non-tied pairwise comparisons. The value of 1.00 means that there are neither winners nor losers. Net Benefit: percentage with which one group has a better win ratio than the other group. Win Odds: probability of a group winning over the other group. ^a^ percentage of pairwise comparisons won by MSS and MSI tumors, respectively. ^b^ number of pairwise comparisons without winners or losers. ^c^ primary event was tumor recurrence; the secondary event was death.

## Data Availability

Data for this study are available at the corresponding author.

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
