# Peer review of "Assessment of the DNA Mismatch Repair System Is Crucial in Colorectal Cancers Necessitating Adjuvant Treatment: A Propensity Score-Matched and Win Ratio Analysis"

_cancers, 2023, doi:10.3390/cancers16010134_

Round 1

Reviewer 1 Report

Comments and Suggestions for Authors

This study evaluated MSI for prognostic use in colorectal cancer. The choice of analyses results in a clear conclusion of the benefit using MSI to optimise treatment in CRC.

Several studies have for decades investigated the role of MSI in CRC as a prognostic factor. Results have been contradictory or at least inconclusive. The aim of this paper was to finally determine the role for MSI in prognosis and therapeutic decisions in CRC. One obstacle has been the fact that MSI tumors constitute less than 15% of all colon cancers why it has been difficult to obtain sufficient numbers for analysis. This paper used a new concept, with two analyses to handle this issue. One new win ratio (WR) analysis, and also a propensity score-matched analysis, and in both these analysis each tumor in the smaller sample set (MSI tumors) are compared to several, up to all, tumors from the other, MSS, cohort. The paper used patients from the same clinic following the same protocol. The important marker, MSI, for analysis was determined using IH or when not conclusive also a PCR based method on tumor DNA.

Finally, 403 patients were included in the study, 351 with MSS tumors and 52 with MSI tumors. Analyses were made for disease-free survival and overall survival and compared between MSS and MSI tumors. All known prognostic factors were confirmed and there was no real difference in prognosis between the two groups of patients who had surgery and no adjuvant treatment. However, in patients who got chemotherapy it was clear that MSS tumors did much better, which is interpreted as the treatment was not beneficial for patients with MSI tumors.

The paper addresses a clinically important subject and the study generated unambiguous results to be used in decisions of treatment in CRC. The paper is well written and clearly describes the different analyses and the strategy to find a conclusive result although sample set was quite small. Adequate figures and detailed tables and figures. The authors describe limitation of small sample set although they have managed to overcome this with the strategy chosen. They also suggest a single recruitment site a limitation – but it is rather the opposite since it facilitates the running of the study following the same protocol.

The conclusion is consistent with the evidence presented in the paper. The paper is well conducted, written and clearly described.

Author Response

Reviewer #1

This study evaluated MSI for prognostic use in colorectal cancer. The choice of analyses results in a clear conclusion of the benefit using MSI to optimize treatment in CRC.

Several studies have for decades investigated the role of MSI in CRC as a prognostic factor. Results have been contradictory or at least inconclusive. The aim of this paper was to finally determine the role for MSI in prognosis and therapeutic decisions in CRC. One obstacle has been the fact that MSI tumors constitute less than 15% of all colon cancers why it has been difficult to obtain sufficient numbers for analysis. This paper used a new concept, with two analyses to handle this issue. One new win ratio (WR) analysis, and also a propensity score-matched analysis, and in both these analysis each tumor in the smaller sample set (MSI tumors) are compared to several, up to all, tumors from the other, MSS, cohort. The paper used patients from the same clinic following the same protocol. The important marker, MSI, for analysis was determined using IH or when not conclusive also a PCR based method on tumor DNA.

Finally, 403 patients were included in the study, 351 with MSS tumors and 52 with MSI tumors. Analyses were made for disease-free survival and overall survival and compared between MSS and MSI tumors. All known prognostic factors were confirmed and there was no real difference in prognosis between the two groups of patients who had surgery and no adjuvant treatment. However, in patients who got chemotherapy it was clear that MSS tumors did much better, which is interpreted as the treatment was not beneficial for patients with MSI tumors.

The paper addresses a clinically important subject and the study generated unambiguous results to be used in decisions of treatment in CRC. The paper is well written and clearly describes the different analyses and the strategy to find a conclusive result although sample set was quite small. Adequate figures and detailed tables and figures. The authors describe limitation of small sample set although they have managed to overcome this with the strategy chosen. They also suggest a single recruitment site a limitation – but it is rather the opposite since it facilitates the running of the study following the same protocol.

The conclusion is consistent with the evidence presented in the paper. The paper is well conducted, written and clearly described.

Reply: We must thank the reviewer who perfectly interpreted our study. The Reviewer has correctly emphasized the statistical methods we applied to overcome limitations due to small number of MSI tumors. The Reviewer, who we still want to thank, stated that the conclusions reported in the study were consistent with the evidence presented in the paper.

Reviewer 2 Report

Comments and Suggestions for Authors

I found this paper by Lieto et al. interesting and clear. The topic is of clinical relevance with respect to the current management patients with colorectal cancer (CRC). The authors investigated the prognostic and predictive of MMR status in a large cohort study.

 I have only minor points to be changed/discussed.

- The format of the tables included in the article should revised in order to improve the readability

- Table 1 and Material and Methods (paragraph “Patients”):

It is not clear the meaning of “Performance Status”. The authors should explain its significance (Main text: material and methods section).

NPS: The authors should explain its significance (Main text: material and methods section).

Postoperative complications: The authors should list the postoperative complications in the Main text: material and methods section.

Figures: It is suggested to improve the resolution/quality of all the figures included in the main text.

Comments on the Quality of English Language

Minor editing of English language required

Author Response

Reviewer #2

I found this paper by Lieto et al. interesting and clear. The topic is of clinical relevance with respect to the current management patients with colorectal cancer (CRC). The authors investigated the prognostic and predictive of MMR status in a large cohort study.

Reply: We thank very much the Reviewer for his/her analysis of our study.

I have only minor points to be changed/discussed.

- The format of the tables included in the article should revised in order to improve the readability. Reply: The Reviewer’s criticism is very correct and we discussed this topic in writing the manuscript. Faced with difficult readability, our aim was to provide the reader with as much information as possible. For this reason we would prefer to leave the tables as currently described. Of course, we could change the tables.

- Table 1 and Material and Methods (paragraph “Patients”):

It is not clear the meaning of “Performance Status”. The authors should explain its significance (Main text: material and methods section)

Reply: Performance Status, according to the Eastern Cooperative Oncology Group (ECOG), was used to evaluate the patient’s status. We have added it to this revised manuscript.

- NPS: The authors should explain its significance (Main text: material and methods section).

Reply: NPS means Naples Prognostic Score; an immune-nutritional score that we proposed in 2017 to identify oncology risk in colorectal cancer patients. Since it, many studies from different Countries in the world have demonstrated the efficacy of this score in several human tumors including lung, pancreas, liver, kidney and stomach cancers. We believe that the reader can easily access the vast bibliography on this topic.

- Postoperative complications: The authors should list the postoperative complications in the Main text: material and methods section.

Reply: The Reviewer’s criticism is correct. Due to limitations in the number of words and paragraphs, we omitted this important issue. Postoperative complications were graded according to the Clavien-Dindo classification (Clavien PA, Barkun J, de Oliveira ML, et al. The Clavien-Dindo classification of surgical complications: five-year experience. Ann Surg 2009;250:187-196). Postoperative complications were defined as grade II or higher of the Clavien-Dindo classification. Now we have added it to the footnotes of the Tables 1-3.

Figures: It is suggested to improve the resolution/quality of all the figures included in the main text.

Reply: Correct, thank you. We have the original files of the figures. During the editing process we can provide these files.

Comments on the Quality of English Language

Minor editing of English language required

Reply: The manuscript has been revised by a native english speaker.

We thank again the Reviewer who helped us to improve our manuscript.